# Analysis of Temperatures Generated during Conventional Laser Irradiation of Root Canals—A Finite Element Study

**DOI:** 10.3390/diagnostics13101757

**Published:** 2023-05-16

**Authors:** Adrian Ștefan Stănuși, Dragoş Laurenţiu Popa, Mihaela Ionescu, Cristian Niky Cumpătă, Gabriel Sebastian Petrescu, Mihaela Jana Ţuculină, Constantin Dăguci, Oana Andreea Diaconu, Lelia Mihaela Gheorghiță, Andreea Stănuşi

**Affiliations:** 1Department of Endodontics, Faculty of Dental Medicine, University of Medicine and Pharmacy of Craiova, 200349 Craiova, Romanialeliagheorghita@yahoo.com (L.M.G.); 2Faculty of Mechanics, University of Craiova, 200349 Craiova, Romania; 3Department of Medical Informatics and Biostatistics, Faculty of Medicine, University of Medicine and Pharmacy of Craiova, 200349 Craiova, Romania; 4Department of Oral and Maxillofacial Surgery, Faculty of Dental Medicine, University Titu Maiorescu of Bucharest, 67A Gheorghe Petrascu Str., 031593 Bucharest, Romania; 5Department of Oral and Maxillofacial Surgery, Faculty of Dental Medicine, University of Medicine and Pharmacy of Craiova, 200349 Craiova, Romania; 6Department of Oro-Dental Prevention, Faculty of Dental Medicine, University of Medicine and Pharmacy of Craiova, 200349 Craiova, Romania; 7Department of Prosthodontics, Faculty of Dental Medicine, University of Medicine and Pharmacy of Craiova, 200349 Craiova, Romania

**Keywords:** finite element analysis, endodontic decontamination, diode laser

## Abstract

The success of endodontic treatment is dependent on the removal of bacteria. A modern strategy to reduce bacterial load is laser irradiation. During this procedure, there is a local increase in temperature with possible side effects. The aim of this study was to determine the thermal behavior of a maxillary first molar when performing the conventional irradiation technique using a diode laser. For this study, a 3D virtual model of a maxillary first molar was created. The preparation of the access cavity, the rotary instrumentation of the palatal root canal and the laser irradiation protocol were simulated. The model was exported in a finite element analysis program where the temperature and heat flux were studied. Temperature and heat flux maps were obtained, and the temperature increase on the internal wall of the root canal was analyzed. The maximum temperature value exceeded 400 °C and was maintained for less than 0.5 s. The obtained temperature maps support the bactericidal effect of diode laser and the limitation of damage to surrounding tissues. On internal root walls, the temperature reached several hundred degrees Celsius, but for very short durations. Conventional laser irradiation is an adjuvant method of decontamination of the endodontic system.

## 1. Introduction

The standard of care for a tooth with pulp and/or periapical lesions is endodontic treatment that ensures its functionality in the entire dento-maxillary apparatus and increases the patient’s quality of life [1]. The success of endodontic treatment is mainly dependent on the removal of bacteria and the smear layer inside the root canals [2,3].

The conventional preparation of the root canal is associated with irrigation with various solutions that have bactericidal action (sodium hypochlorite, EDTA) and that require direct contact with the dentine surface.

Modern techniques for mechanical instrumentation of root canals involve the use of endodontic files with a large taper and present various risks, such as perforations and stripping. At the same time, by using these endodontic techniques, the walls of the root canals are prepared in a less conservative manner, which leads to the reduction of the mechanical resistance of the root walls and may compromise the future restoration of the teeth in question. However, these techniques help to remove debris and improve the access of irrigants in order to disinfect the endodontic system [4,5,6,7].

The most used solution to irrigate root canals is sodium hypochlorite, which dissolves organic matter and effectively kills microorganisms. The best effects are obtained when using higher concentrations of the hypochlorite solution (4% or 5.25%). Along with sodium hypochlorite, EDTA is used to irrigate the root canal, which removes the smear layer. Agitation of these solutions by sonic or ultrasonic activation can improve their action [8,9].

Knowing that these irrigants have a limited penetration into the dentinal canals—a maximum of 100 µm—while bacteria can be identified at 1000 µm in depth [10,11,12] and that the instrumentation of the root canals cannot be carried out in all ramifications of the endodontic system, it becomes imperative to introduce in the endodontic treatment protocol a root canal decontamination technique to inhibit the development of a possible recurrent infection [10].

A modern strategy to reduce the bacterial load in the root canal is represented by irradiation using dental lasers [2]. There are numerous studies that demonstrate the effectiveness of these medical devices [1,2,3,4,5,6,7,8,9,10,11,12,13,14,15,16,17].

The reduction of the bacterial load by the action of the laser light is explained by two mechanisms. The first mechanism refers to the absorption of laser light by bacteria with its direct destruction. This phenomenon is directly dependent on the existence of black pigments (protoporphyrin IX) that can absorb a limited portion of laser wavelengths. The second mechanism refers to the absorption of laser light in the dentinal substrate on which the bacteria adhere, with a local increase in temperature that leads to the death of the attached microorganisms [18]. Thermographic studies have re-presented a step forward in identifying safe parameters for laser decontamination of root canals [19].

Dental lasers can be used for direct irradiation of the dentine walls or for the irradiation/activation of photoactive substances/irrigants, having indirect clinical action on the endodontic system [19].

The activation protocol of endodontic irrigators with lasers was described for the first time in 2008 [20,21]. Since then, studies that document the advantages of these techniques have been made. The studies carried out in the field of laser-assisted endodontics showed that, by associating laser irradiation with various irrigation solutions (EDTA, sodium hypochlorite, peroxide), the cleaning of the endodontic system was improved, the amount of biofilm removed was greater and antibacterial properties were superior [8,22,23,24]. Certain specialists stated that the ideal protocol for disinfection of the endodontic system is carried out by irrigating the root canals with 5.25% sodium hypochlorite, in association with laser irradiation, obtaining superior bactericidal effects, including on Enterococcus faecalis [13].

The laser technique used since the beginning involves the use of fibers or tips with terminal emission (“end firing”) positioned in the root canal, usually 1 mm away from the apex. This technique is suitable for most wavelengths in the visible (445 and 532 nm), near-infrared (from 810 to 1340 nm) and medium-infrared (2780 and 2940 nm) areas of the electromagnetic spectrum. The activation of the laser starts when the fiber/tip is withdrawn in a helical/circular movement over a certain time interval (usually 2 mm/s) [19].

During dental laser irradiation of the root canal, there is a local increase in temperature with the possibility of developing unwanted side effects, especially if the practitioner does not follow the protocol and does not adapt the laser power to the clinical situation [25,26]. The increase in temperature and the development of side effects are dependent on certain elements related to the morphology of the root canal and the physical and chemical properties of the dental and periapical tissues, but also on elements related to the settings of the laser device and the irradiation protocol used [26,27].

The aim of this study was to determine the thermal behavior of a maxillary first molar when performing conventional diode laser irradiation of the palatal root canal, using the Finite Element Method to obtain a 3D virtual model and simulate the clinical situation. Accurate recording of temperatures generated in the form of temperature maps for the main components of the 3D virtual model was intended.

## 2. Materials and Methods

### 2.1. Obtaining the Virtual 3D Model of the Maxillary First Molar

In order to create a 3D virtual model of a maxillary first molar, a set of CBCT tomo-graphic images of a 30-year-old patient was used. The patient presented herself at the Endodontics Clinic of the Faculty of Dental Medicine, UMF Craiova, because she had persistent symptoms indicating an endodontic periapical lesion, although the endodontic treatment was performed by a specialist. The CBCT tomographic investigation was conducted in order to evaluate the endodontic treatment and to evaluate the periapical tissues.

The patient expressed her consent in writing for the use and publication of the data, and the study obtained the approval of the Ethics Commission of the University of Medicine and Pharmacy from Craiova no. 28/24.02.2021. The CBCT images were transformed into virtual geometries using initially the InVesalius program (Figure 1).

The geometric structure of the maxillary first molar was obtained in the Geomagic program through different removal operations (Figure 2).

Using techniques and methods specific to reverse engineering, the initial model was processed (Figure 3a) and then transferred and transformed into a virtual solid in the SolidWorks program (Figure 3b). To obtain the dentine component of the 3D virtual model, the outer surface of the molar was used and an off-set surface was created in Geomagic using specific reverse engineering methods and techniques (Figure 3c).

To obtain the component represented by the endodontic system of the maxillary first molar, the pulp chamber and the 4 main root canals, first a primary geometry was generated in SolidWorks using simple shapes, CAD methods and techniques (Figure 4a,b).

This primary model was exported to Geomagic where it underwent complex editing and transformation operations (Figure 4c,d). To obtain the final form, a series of specialized works were analyzed [28,29]. Afterwards, the model was exported to SolidWorks and was transformed into a virtual solid (Figure 4e,f).

To obtain the component represented in the periodontal space, the initial model of the molar was imported into Geomagic, where it was subjected to an off-set surface opera-tion at a distance of 0.2 mm [30]. This model was exported to SolidWorks and an internal cavity was obtained at the same distance of 0.2 mm (Figure 5).

To obtain the model of the alveolar bone of the maxillary first molar, the Bone filter was used in the InVesalius program, and the model was imported into Geomagic for editing and processing (Figure 6).

The components of the dental assembly thus obtained were loaded into the SolidWorks program (Figure 7).

### 2.2. Simulation of the Rotary Instrumentation of the Palatal Root Canal

The access cavity was created based on recommendations from the literature [29] (Figure 8).

To be able to simulate the rotary instrumentation of the palatal root canal of the maxillary first molar with the Reciproc Blue system, a virtual 3D model of an object was de-fined to possess the taper of an R50 ISO 50 taper 0.05 needle and follow the root canal morphology. Thus, the model in Figure 9 was obtained, which was inserted into the virtual 3D model of the maxillary first molar and used to generate a cavity in the studied root canal.

### 2.3. Assignation of Physical Characteristics

The model of the maxillary first molar was imported into Ansys Workbench, a program that allows the analysis of mechanical or thermal behavior using the finite element method (Figure 10). In the Engineering Data module, the tissues specific to the studied model and their physico-thermal characteristics (Table 1) were added, as known from the literature [31,32,33]. The division of the model into finite elements followed (Figure 11).

### 2.4. Simulation of Conventional Laser Irradiation

It was considered that the elements of the obtained 3D virtual model have a temperature of 37 °C (Figure 12).

The conventional irradiation of the palatal root canal of the maxillary first molar was imagined using the diode laser with a wavelength of 940 nm at a power of 1 W and the endodontic tip of 200 µm. The protocol involved making a helical movement with the tip inside the canal with a speed of 2 mm/s. In the thermal analysis, a helical movement of the heat source could not be created, and the following simplifications were made to define heat fluxes:The sectors inside the root canal are 1 mm high and would be covered in 0.5 s. The heat flux, defined in Ansys, will act on each sector for 0.5 s.The diode laser operates with a power of 1 W during irradiation protocol. This power will be related to the surface of each sector, successively.

To identify the irradiated sector when the laser tip moves along the root canal, the model of the prepared root canal in SolidWorks was transposed in three projection planes of the coordinate system (Figure 13a). The graphic construction that helped determine the irradiated sector started from drawing the diameter of the spatial circle that delimits the first sector with blue color. Perpendicular to the diameter, a line segment was drawn that simulates the direction of the laser beam when the tip is in position 1 with red color (Figure 13b). It is found that, in this case, sector 1 is the irradiated one. The construction is repeated for the entire root canal to determine the irradiated sectors and each sector and the tip position are noted (Figure 13c).

Table 2 describes the successive position of the tips, the irradiated sectors of the root canal and the duration of the laser action for a single work cycle. The complete treatment protocol was considered to consist of 3 work cycles, interrupted by 10-s breaks (Table 3).

### 2.5. Calculation of Heat Flux

To determine the heat fluxes or power densities (Prldensity), the laser power (P = 1 W) was related to the area of each sector (Arl). Using CAD measurement techniques, the sur-faces for the targeted sectors of the analyzed root canal were determined (the surfaces are shown in green), as shown in Figure 14. The data obtained were introduced in Table 4.

At that time, there were all the necessary data to define the heat fluxes (power densities) on the analyzed virtual model. The selected surfaces for each flux were the sectors previously determined by geometric analysis. Figure 15 shows the positions of these fluxes in Ansys Workbench and the temporal variation of the flux acting on a sector.

A physical phenomenon that was considered is thermal convection [34,35,36]. The studied model is mainly affected by the phenomenon of free convection caused by partial re-lease of heat. In this case, the value of convection is between the values of 2.5–25 W/m^2^ · K. For the simulation of the analyzed model, an average value of 13.25 W/m^2^ · K was chosen, and the surfaces affected by this phenomenon were the sectors in the root canal, but also the access cavity (Figure 16). The model was prepared for the simulation run, thus having all the data and constraints defined.

## 3. Results

After running the application, maps of temperatures and heat fluxes were obtained. Figure 17 shows the initial temperature map and the initial map of the total heat flux. The Ansys Workbench program allows the obtaining of a dynamic temperature map, similar to a movie of the simulation. A sequence of this film is shown in Figure 18, after using a sectioning plane that crosses the 3D virtual model through the palatal root canal.

A number of results can be exported and analyzed in Microsoft Office Excel files. Thus, the global temperature values are shown in Figure 19. The maximum temperature recorded was 427.09 °C and was maintained for a maximum of 0.5 s. It was recorded in seconds 2, 19 and 36 of the protocol.

Additionally, the Ansys program allows placing samples from which the temperature can be read during the simulation. Thus, for a sample placed in sector 1 of the root canal, on the internal wall (Figure 20), the diagram from Figure 21 was determined. The maximum temperature recorded for this sample was 234.35 °C and was maintained for a maximum of 0.5 s. This temperature was recorded in seconds 18 and 35 of the protocol.

Similarly, a temperature diagram was obtained for a sample positioned in sector 7, on the internal root wall, as shown in Figure 22 and Figure 23. The maximum temperature recorded for this sample was 367.58 °C and was maintained for a maximum of 0.5 s. This temperature was recorded in seconds 4, 21 and 38 of the protocol.

A sample was placed in the apical third of the root canal, on the external surface, and the diagram in Figure 24 was obtained. The maximum temperature recorded for this sample was 132.51 °C and was maintained for a maximum of 0.5 s. This temperature was recorded in seconds 18 and 35 of the protocol.

The temperature was also checked on the periodontal space structure in the vicinity of the root apical third and it was observed that it remains constant at the physiological value, obtaining the diagram in Figure 25. Finally, the temperature was also checked on the alveolar bone structure corresponding to the root apical third (Figure 26).

## 4. Discussion

One of the most recent advances in endodontics is the use of the dental laser as an adjunctive therapy to reduce the bacterial load in the endodontic system. This has the great advantage of reaching areas that are not accessible to conventional irrigation solutions [37].

Research in the field has demonstrated that all types of dental lasers, using different power levels, are capable of destroying the pathogens responsible for endodontic infections [1,10,11,38,39,40,41,42,43,44,45]. This bactericidal action is due to the photo-thermal effect that causes the destruction of the cell wall and the modification of the osmotic gradient with cell death [11,46].

Recent studies have focused on unwanted side effects that may occur during laser irradiation of root canals. The local increase in temperature during the irradiation of root canals with dental lasers is a topic that has particularly attracted the attention of specialists [24,25,33,42,46,47,48,49,50,51,52,53,54], but the methods used so far for recording the temperature may present different limitations depending on the materials and the working method used.

By inserting the teeth into various materials with similar, but not identical, thermal properties to the oral tissues, the temperature generated and dissipated in the dental tissues, the periodontal ligament and the alveolar bone cannot be correctly quantified. In addition, the use of thermocouples and thermographic cameras to record the temperature during the operative protocol does not allow analysis of the temperature variation on the internal root wall, at any time during the intervention or in any other area. From the data published so far in the specialized literature, we have not found an in vino study in which the temperatures generated by the laser irradiation of the endodontic system were at the level of the internal and/or external root walls.

It must also be emphasized that the results of research regarding the increase in temperature during the laser irradiation of root canals differ depending on the type of laser used, the set power and the working protocol.

Finite element analysis is used in numerous studies related to stress and temperature distribution, being superior to thermocouples and infrared cameras. This working method is particularly important when analyzing a tooth with a complex root anatomy. The finite element method allows accurate temperature measurement at any time and at any location in the dental tissues during the simulated working protocol [55].

The finite element method and the thermal study carried out in this research allowed the simulation of the conventional laser irradiation protocol of the palatal root canal of a maxillary first molar and the exact recording of the generated temperatures. The 3D virtual model obtained presented the specific internal and external morphology of the selected tooth, as well as the physical-thermal characteristics of the tissues.

Due to the fact that through the finite element method all this information can be introduced, respecting the characteristics of natural tissues, as they are known, we are confident that the results obtained have a high degree of similarity with those that can be obtained by applying this technique in vivo.

In the presented study, it was possible to measure the temperature at the level of the internal and external root wall during the irradiation protocol by analyzing the thermal maps for samples positioned at that level. The temperature recorded on the external root surface in the apical third exceeded 100 °C, and on the internal root surface it exceeded 200 °C. Similar results were obtained by Gutknecht in 2005 [33]. He observed a rapid decrease in temperature as the laser tip was removed, an aspect also recorded in this study. These temperatures were maintained for a maximum of 0.5 s. Gutknecht stated [33] that the high temperatures recorded cannot cause the root walls to melt or recrystallize but can affect the microorganisms in that area.

Compliance with the protocol of root canal irradiation is important to avoid side effects. Maintaining these high temperatures for a maximum of 0.5 s is dependent on respecting a tip removal speed of 2 mm/s from the root canal.

In the presented study, there was no increase in temperature at the alveolar bone and the periodontal ligament, which could cause irreversible changes. It is considered that under these conditions the surrounding tissues cannot be affected by the heat generated during the irradiation of the root canal.

Numerous studies have demonstrated that the bactericidal effect of the diode laser is due to this increase in local temperature [1,2,3,13,56], which ensures the reduction of the bacterial load in the main and secondary root canals, which are inaccessible to conventional preparation [33]. In addition, bacteria cannot develop resistance to the action of the laser [13].

The research presented allowed the simulation of conventional irradiation using a diode laser of the palatal root canal of a maxillary first molar and the analysis of the temperatures generated in the dental and surrounding tissues. The finite element method was useful for the simulation of the irradiation protocol and for the analysis of the generated temperatures.

The simulation carried out provides support for in vitro studies that analyze the bactericidal effect of the increase in temperature at the level of the root walls. The obtained temperature maps support the bactericidal effect of the diode laser through photo-thermal action and the limitation of damage to the surrounding tissues under the conditions of compliance with the work protocol.

In addition to previous studies in this domain, this research highlights the temperature recorded on the internal root wall at any time during the conventional laser irradiation protocol of the root canal.

The weak point of this study is represented by the fact that the laser irradiation protocol was simulated for a single root canal, prepared using a single system of endodontic files.

In our future research, we want to carry out a similar study on teeth with smaller roots using various root canal preparation files with more or less conservative techniques and various endodontic irrigation solutions in order to correlate the temperatures obtained with possible thermal effects and with the thickness of the root walls. Additionally, another future research direction will be represented by a study that will address the identification of potential thermal alteration lesions at the level of the internal root wall using a Scanning Electron Microscope.

## 5. Conclusions

The finite element method can be used to create a 3D virtual model of the maxillary first molar and to simulate the rotary instrumentation of the root canals and conventional laser irradiation.

Using the finite element method, it is possible to analyze the temperatures generated during the laser irradiation of the root canals.

The temperature of the alveolar bone and the periodontal ligament did not change during the irradiation protocol; therefore, thermal injuries will not occur at the alveolar bone and the periodontal ligament if the protocol is respected.

The other structures of the 3D virtual model reached high temperatures, of several hundred degrees Celsius, but for very short periods of time, under 0.5 s.

The maximum values of the recorded temperatures were located on the internal root wall, in the vicinity of the tip position of the dental laser.

If this working protocol for root canal irradiation is followed, no thermal injuries will occur on the external root walls or the surrounding periodontal tissues. The possibility of thermal injuries developing on the internal root walls is an aspect that must be further investigated.

Conventional laser irradiation of root canals is an adjuvant method of decontamination of the endodontic system through the thermal effect on bacterial cells and on the substrate they adhere to.

## Figures and Tables

**Figure 1 diagnostics-13-01757-f001:**
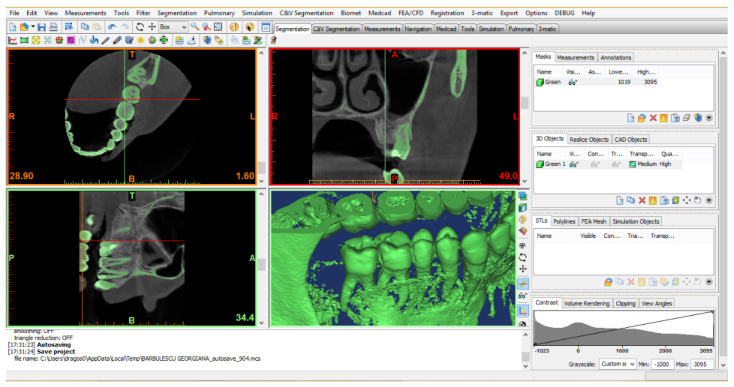
The interface of the InVesalius program.

**Figure 2 diagnostics-13-01757-f002:**
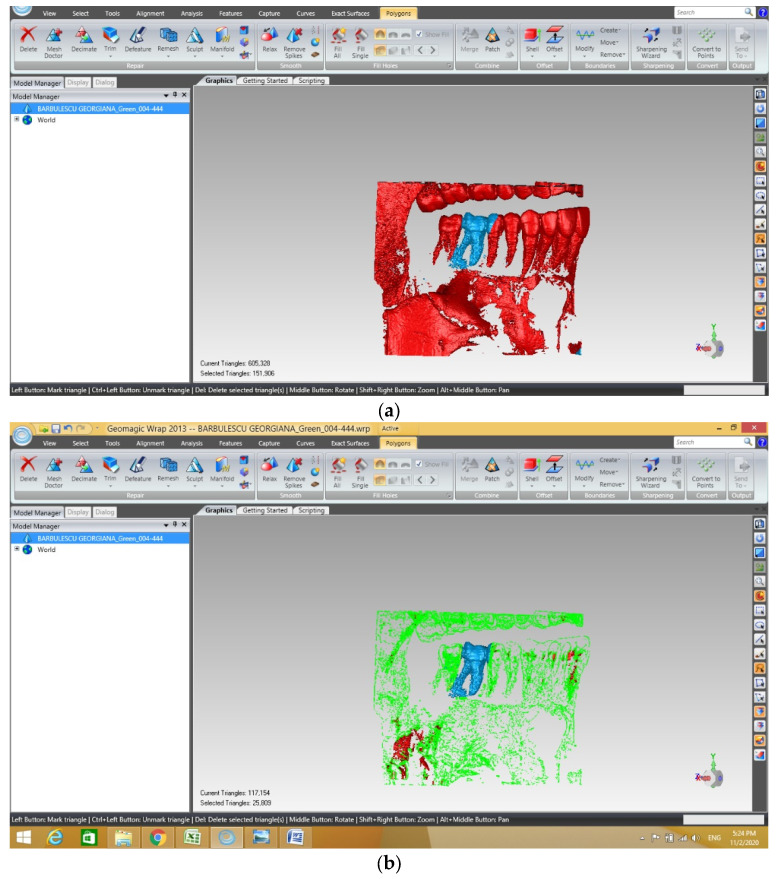
Removal steps in Geomagic. (**a**) selection of tissues to be removed marked with red; (**b**) removal of tissues selected.

**Figure 3 diagnostics-13-01757-f003:**
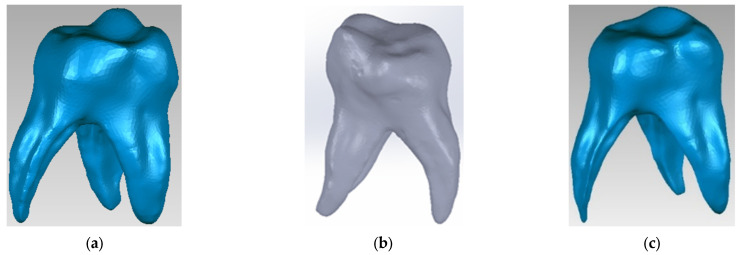
(**a**) Maxillary first molar initial model in Geomagic; (**b**) Maxillary first molar initial model in SolidWorks; (**c**) Dentine component in Geomagic.

**Figure 4 diagnostics-13-01757-f004:**
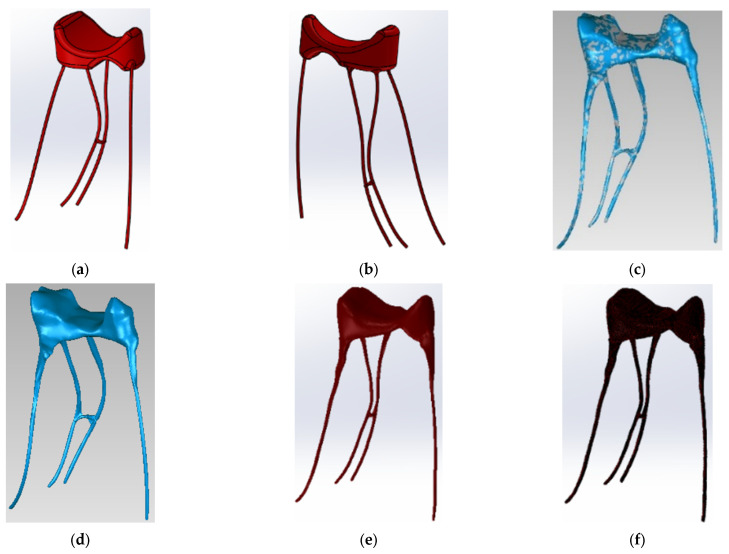
(**a**,**b**) Primary model of the endodontic system in SolidWorks; (**c**,**d**) Operations in Geomagic to obtain the model of the endodontic system; (**e**,**f**) Model of endodontic system in SolidWorks.

**Figure 5 diagnostics-13-01757-f005:**
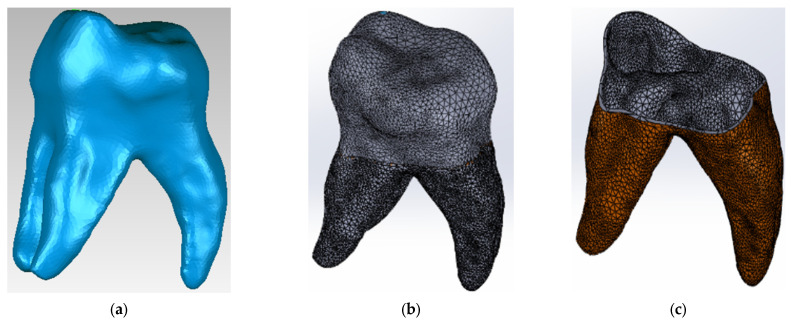
Stages of defining the periodontal space. (**a**) initial model of the molar; (**b**) model of the molar imported in Geomagic; (**c**) model of the periodontal space in Geomagic.

**Figure 6 diagnostics-13-01757-f006:**
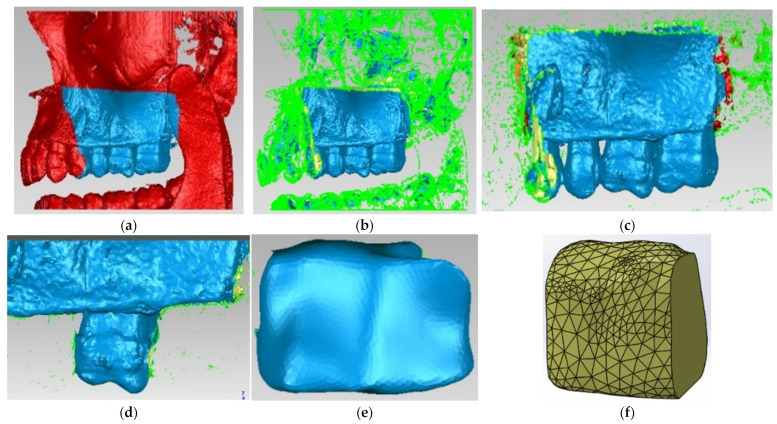
Stages of obtaining the model of the alveolar bone. (**a**) selection of unwanted tissues; (**b**) removal of selected tissues; (**c**) view after removing of unwanted structures; (**d**) model of molar with alveolar bone; (**e**) model of alveolar bone in InVaesalius; (**f**) model of alveolar bone imported in Geomagic.

**Figure 7 diagnostics-13-01757-f007:**
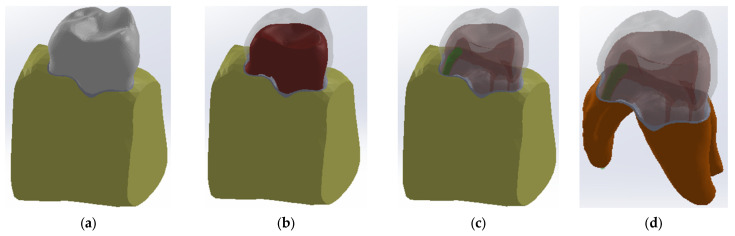
The primary model of the dental system analyzed. (**a**) no transparency; (**b**) inhanced transparency in enamel to view dentine marked with red; (**c**) inhanced transparency in enamel and dentine to view endodontic space marked with dark red (**d**).

**Figure 8 diagnostics-13-01757-f008:**
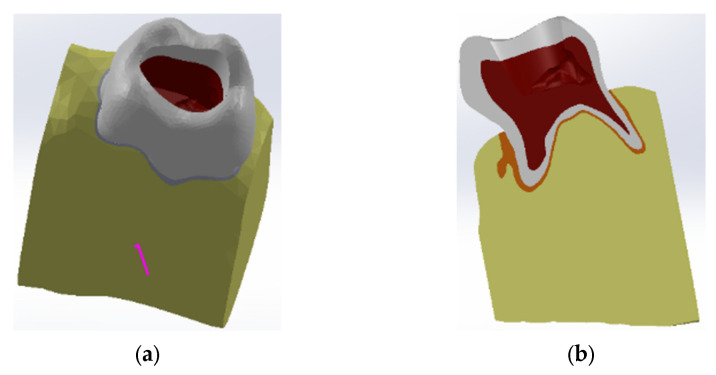
Maxillary first molar model with the access cavity. (**a**) view from the occlusal surface; (**b**) longitudinal section.

**Figure 9 diagnostics-13-01757-f009:**
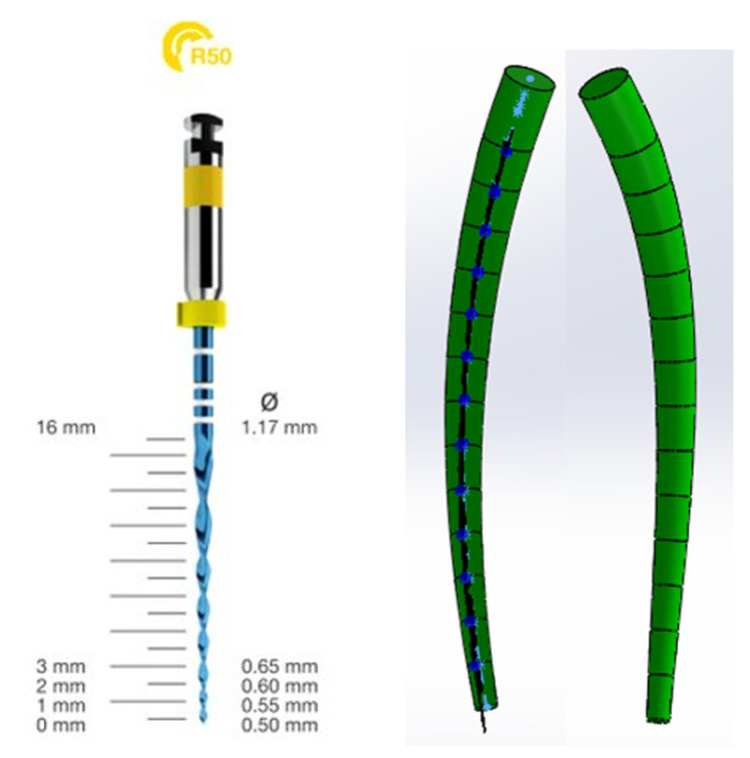
Virtual model of rotary root canal instrumentation.

**Figure 10 diagnostics-13-01757-f010:**
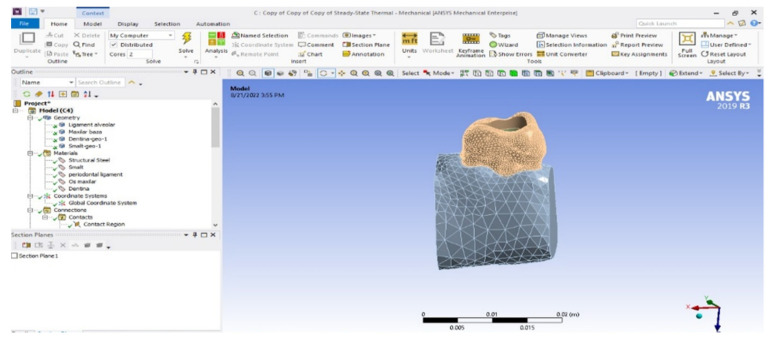
Model analyzed in Ansys Workbench.

**Figure 11 diagnostics-13-01757-f011:**
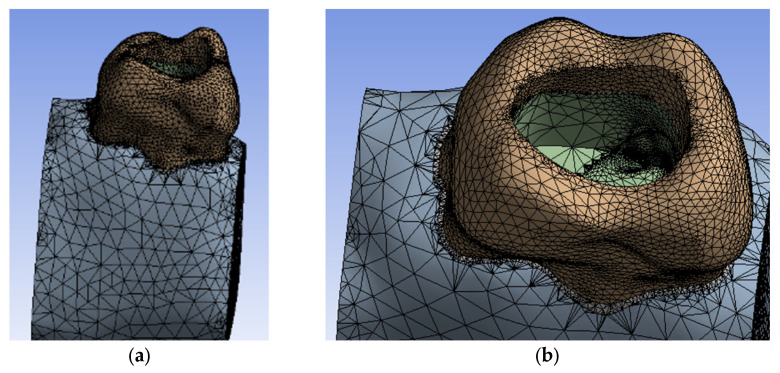
Finite element structure of the analyzed system. (**a**) view from proximal surface; (**b**) view from the occlusal surface.

**Figure 12 diagnostics-13-01757-f012:**
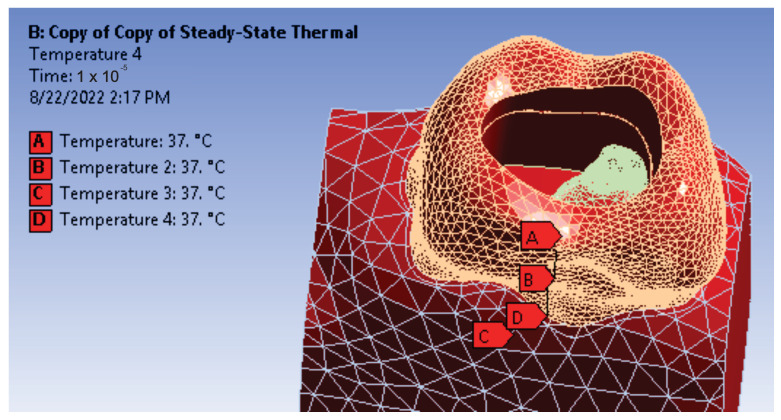
Definition of temperature sources.

**Figure 13 diagnostics-13-01757-f013:**
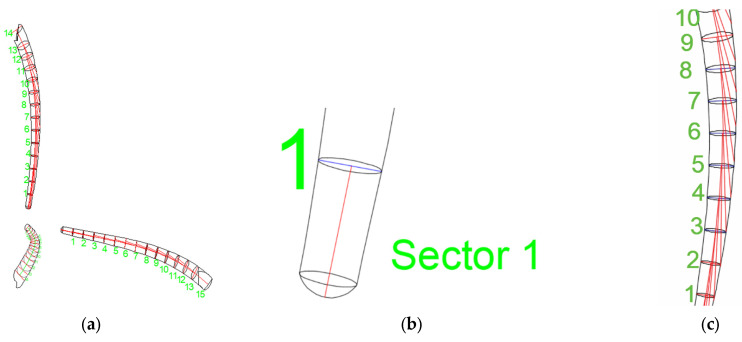
Graphic construction of tip position and irradiated sectors marked by numbers from 1 to 14. (**a**) root canal in 3 planes; (**b**) tip position in sector 1; (**c**) tip position in root canal sectors from sector 1 to sector 10.

**Figure 14 diagnostics-13-01757-f014:**
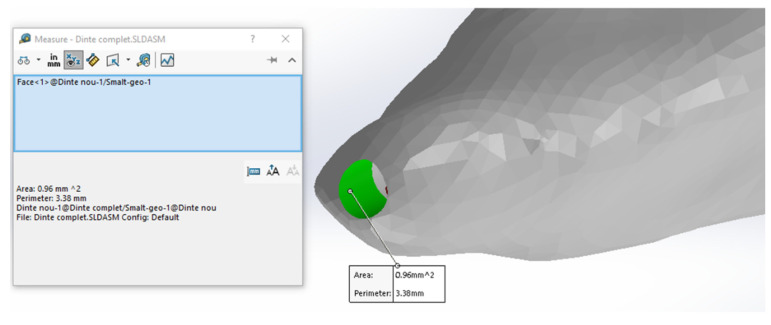
Measuring the areas of sectors.

**Figure 15 diagnostics-13-01757-f015:**
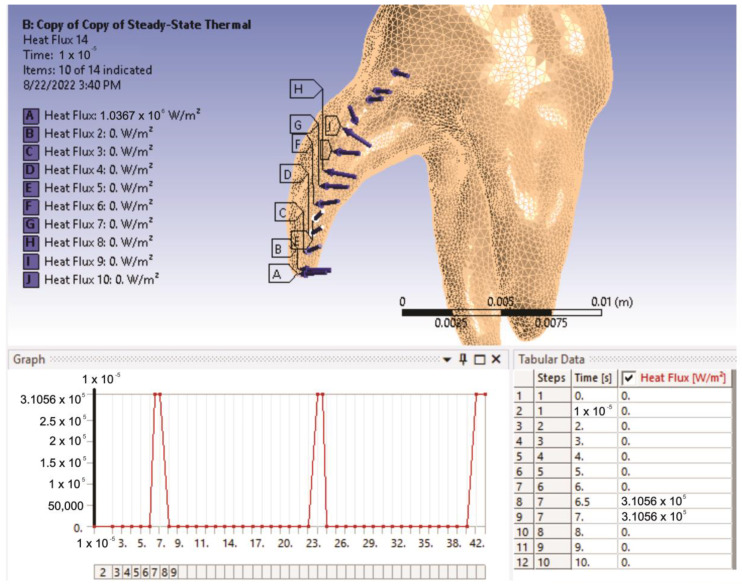
The position of heat fluxes in the analyzed model.

**Figure 16 diagnostics-13-01757-f016:**
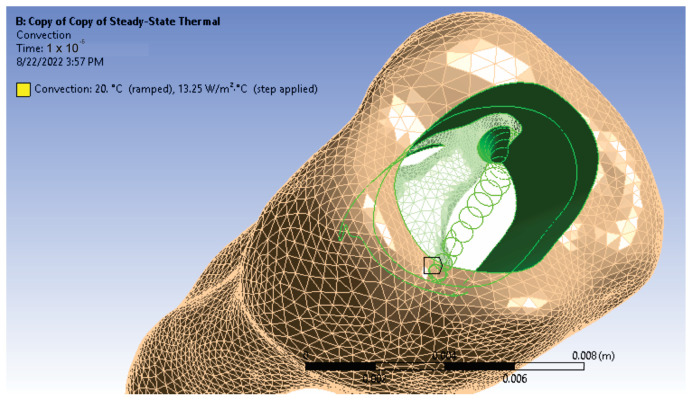
Surfaces subject to free thermal convection.

**Figure 17 diagnostics-13-01757-f017:**
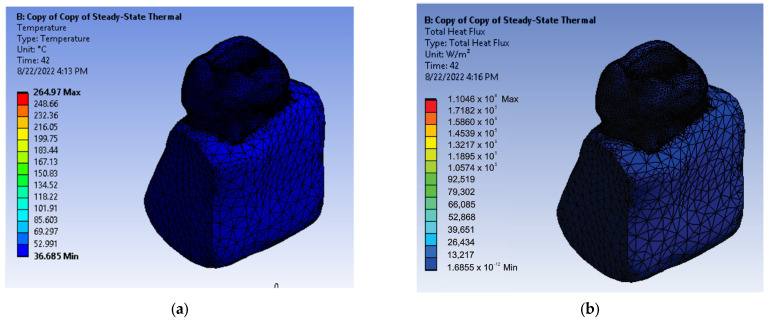
(**a**) Initial temperature map; (**b**) initial map of the total heat flux.

**Figure 18 diagnostics-13-01757-f018:**
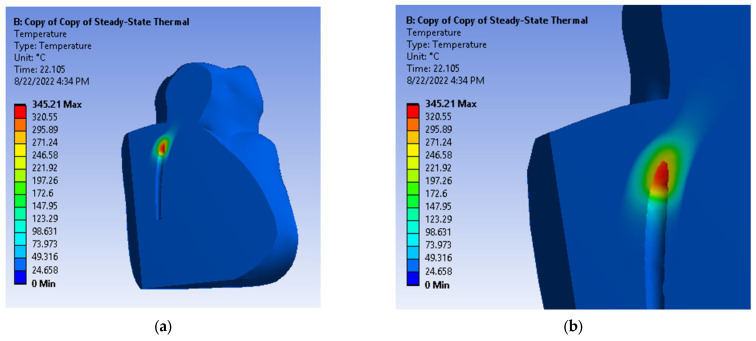
Temperature maps after sectioning. (**a**) view of the irradiated palatal root canal from proximal surface; (**b**) closer view of the irradiated palatal root canal from proximal surface.

**Figure 19 diagnostics-13-01757-f019:**
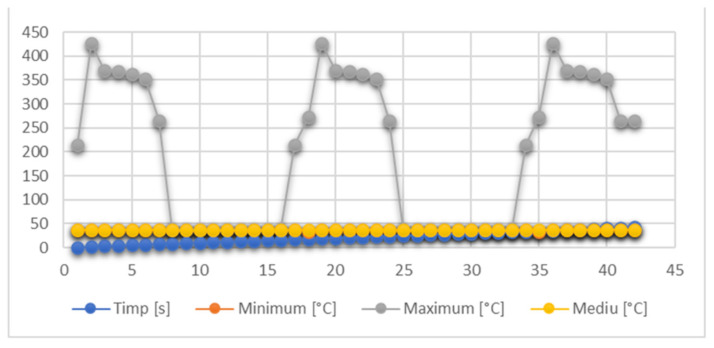
Global temperatures.

**Figure 20 diagnostics-13-01757-f020:**
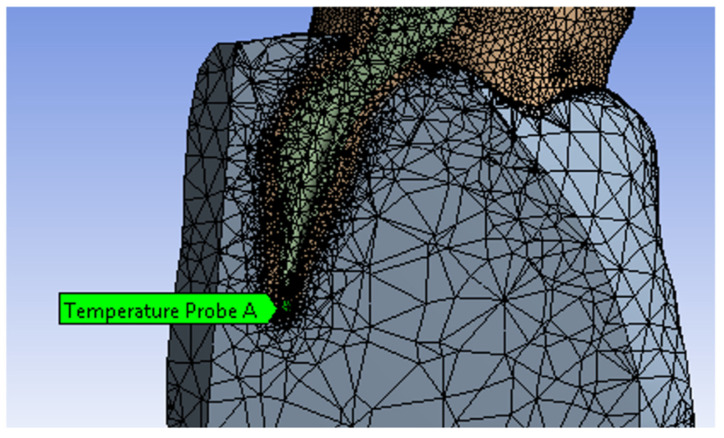
Sample position in sector 1.

**Figure 21 diagnostics-13-01757-f021:**
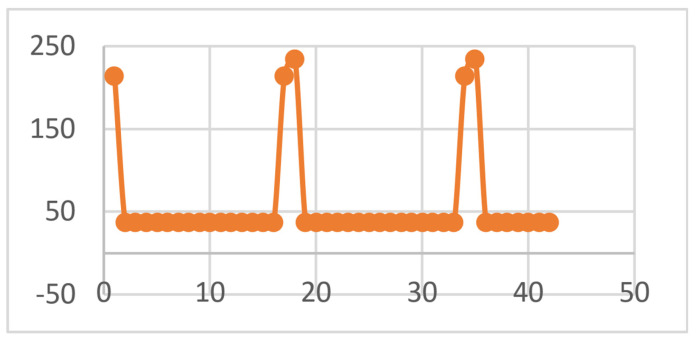
Diagram of sector 1 temperatures.

**Figure 22 diagnostics-13-01757-f022:**
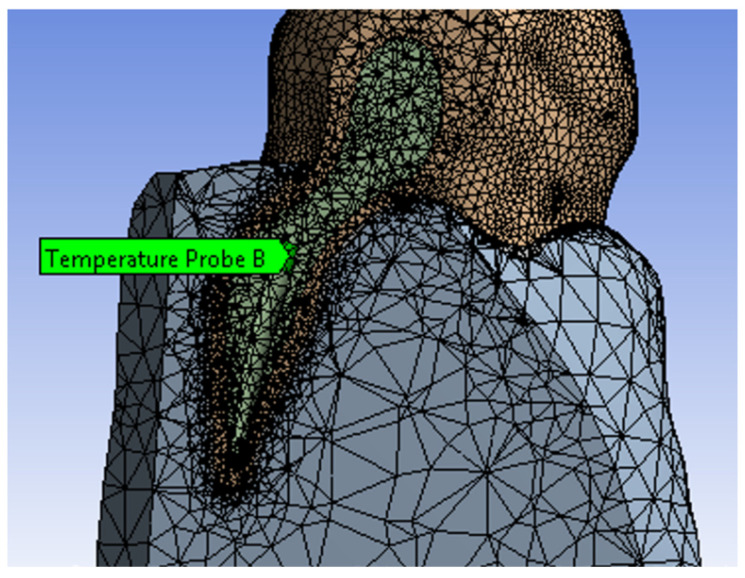
Sample position in sector 7.

**Figure 23 diagnostics-13-01757-f023:**
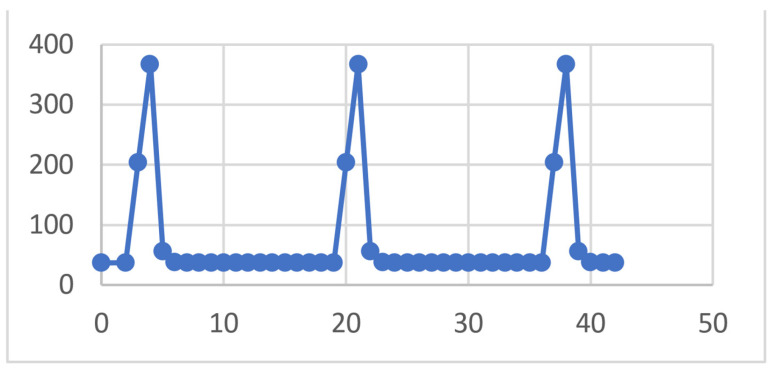
Diagram of sector 7 temperatures.

**Figure 24 diagnostics-13-01757-f024:**
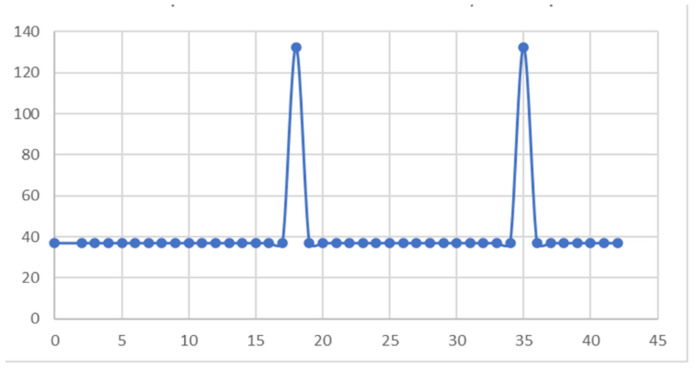
Diagram of temperatures on the external root surface in the apical third.

**Figure 25 diagnostics-13-01757-f025:**
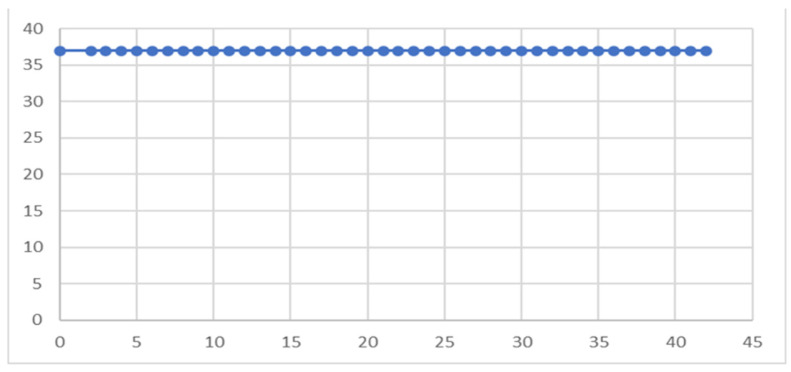
Periodontal ligament temperature.

**Figure 26 diagnostics-13-01757-f026:**
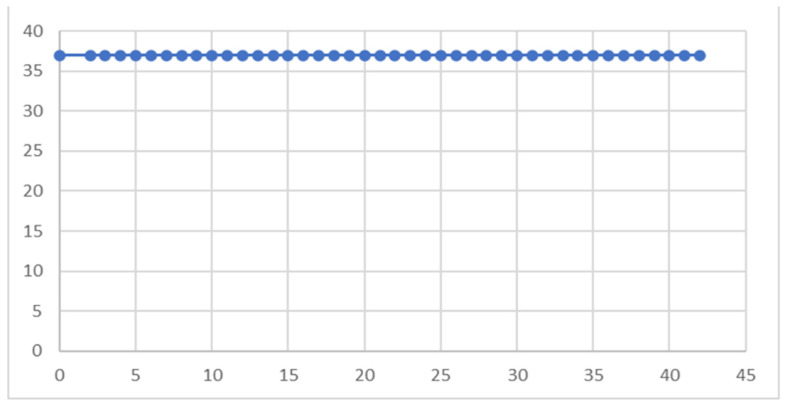
Alveolar process temperature.

**Table 1 diagnostics-13-01757-t001:** Physico-thermal characteristics of dental tissues.

Virtual Model Component	Tissue	Density [kg/m^3^]	Thermal Conductivity [W/m·°C]	Mass Heat Capacity [J/kg·°C]
Enamel structure	Enamel	2958	0.93	710
Dentine structure	Dentine	2140	0.58	1590
Periodontal ligament structure	Periodontal ligament	1100	0.58	4820
Alveolar bone structure	Alveolar bone	2310	1	2650

**Table 2 diagnostics-13-01757-t002:** Tips positions and irradiated sectors.

Tips Positions	Irradiated Sector	Duration of Action [s]
1	Sector 1	0–0.5
2	Sector 1	0.5–1
3	Sector 1	1–1.5
4	Sector 1	1.5–2
5	Sector 2	2–2.5
6	Sector 3	2.5–3
7	Sector 4	3–3.5
8	Sector 5	3.5–4
9	Sector 6	4–4.5
10	Sector 7	4.5–5
11	Sector 8	5–5.5
12	Sector 9	5.5–6
13	Sector 10	6–6.5
14	Sector 11	6.5–7

**Table 3 diagnostics-13-01757-t003:** Complete root canal irradiation protocol.

Duration of Action [s]	Type of Action
0.0–7.0	Laser irradiation
7.0–17.0	Pause
17.0–24.0	Laser irradiation
24.0–34.0	Pause
34.0–41.0	Laser irradiation

**Table 4 diagnostics-13-01757-t004:** Values of heat fluxes for each sector.

Sector	Sector Area [mm^2^]	The Value of the Heat Flux [W/m^2^]	Duration of Treatment Cycle [s]
1	0.96463	1,036,667	0–0.5
1	0.96463	1,036,667	0.5–1
1	0.96463	1,036,667	1–1.5
1	0.96463	1,036,667	1.5–2
2	1.63	613,496.9	2–2.5
3	1.81	552,486.2	2.5–3
4	2.12	471,698.1	3–3.5
5	2.28	438,596.5	3.5–4
6	2.43	411,522.6	4–4.5
7	2.6	384,615.4	4.5–5
8	2.75	363,636.4	5–5.5
9	2.91	343,642.6	5.5–6
10	3.07	325,732.9	6–6.5
11	3.22	310,559	6.5–7

## Data Availability

The authors declare that the data from this research are available from the corresponding authors upon reasonable request.

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
