# Peer review of "Analysis of Temperatures Generated during Conventional Laser Irradiation of Root Canals—A Finite Element Study"

_diagnostics, 2023, doi:10.3390/diagnostics13101757_

Round 1

Reviewer 1 Report

Dear Authors, 

the work done is of an excellent standard. I believe that the methodology is confirmed, and that some aspects related to daily clinical practice, future in vivo applications and various evaluations of their clinical use can be improved.

The paper is a finite element study on the analysis of temperatures generated during conventional laser irradiation of root canals.

The Authors made a great work in terms of methodology and the paper sounds scientific and well written.

However, some improvements are mandatory before acceptance.

The abstract is well written, complete and summary in its various aspects. The keywords are complete and appropriate.

In the abstract:

·        “4000C and was maintained for less than 0.5 sec.” please check if the measurement units are reported correctly.

In the Introduction:

·        I believe that the introduction should be slightly increased with some assessments on the need to shape the canals in a more or less conservative way to achieve this result, and on the type of irrigant present inserted inside the endodontic system to achieve the best possible disinfection by activation with Laser. As is clear, the greater the canal shape, the better the removal of debris and the possibility of disinfecting through the use of different irrigants, but the greater the intraoperative risks of stripping, perforations and intracanal separation of the instruments, as well as a reduced resistance of the entire tooth/restoration system that will be placed.

“Gambarini G, Miccoli G, D'Angelo M, Seracchiani M, Obino FV, Reda R, Testarelli L. The relevance of operative torque and torsional resistance of nickel-titanium rotary instruments: A preliminary clinical investigation. Saudi Endod J 2020;10:260-4”

Materials and methods are clear and well explained:

·        Why was a 3d diagnostic exam prescribed? Please explain more about how this exam was obtained.

Different aspects are analyzed with a dedicated statistical test. The authors did a great job in the explication of all the variables identified and included in the study.

Results are easy to understand and comprehensive. All the studied characteristics were reported in tables which are clear and concise.

In the discussion:

·        “The local increase in temperature during the irradiation of root canals with dental lasers is a topic that has particularly attracted the attention of specialists [13, 21, 31, 35-44], but the methods used so far for recording the temperature may present different limitations depending on the materials and the working method used.” how do the authors believe that temperature measurement could be different between finite element simulation and in vivo application of such a technique?

Discussion: The overall is comprehensive, concise and complete in its various aspects.

Conclusions are concise and clear.

Figures and labels are clear and easy to comprehend.

English is clear and easy to understand.

Bibliography should be formatted respecting the journal’s requirements and no improper citations are evidenced.

 Author Response

Thank you for the evaluation and for the recommendations!

Point 1: In the abstract:·        “4000C and was maintained for less than 0.5 sec.” please check if the measurement units are reported correctly. –

Response 1:  It was a typo that we corrected. Instead of 4000C we wrote 4000C in the revised form of the article.

Point 2: In the Introduction:

  • I believe that the introduction should be slightly increased with some assessments on the need to shape the canals in a more or less conservative way to achieve this result, and on the type of irrigant present inserted inside the endodontic system to achieve the best possible disinfection by activation with Laser.

    “Gambarini G, Miccoli G, D'Angelo M, Seracchiani M, Obino FV, Reda R, Testarelli L. The relevance of operative torque and torsional resistance of nickel-titanium rotary instruments: A preliminary clinical investigation. Saudi Endod J 2020;10:260-4”

Response 2: The introduction was improved according to the recommendations, by adding information about: the more or less conservative way of preparing root canals, the types of irrigation solutions used and the optimal protocol for decontamination of the endodontic system by laser-assisted techniques.

Point 3:  Why was a 3d diagnostic exam prescribed? Please explain more about how this exam was obtained.

Response 3: Materials and methods have been improved with the reasoning of the 3D exam.

Point 4:  how do the authors believe that temperature measurement could be different between finite element simulation and in vivo application of such a technique?

Response 4: The authors consider that due to the high degree of precision we obtained by characterizing all the tissues involved, regarding the morphological aspects and the physical and thermal properties, the measurements obtained do not differ significantly from those obtained during the in vivo application of this technique.

Point 5. Bibliography should be formatted respecting the journal’s requirements

Response 5: Bibliography has been improved as recommended.

Reviewer 2 Report

Dear Authors,

I'm sorry, but in my opinion, your work can't be accepted because the irrigation technique shown can be clinically dangerous:

You wrote:

"Thermal injuries should not occur at the alveolar bone and the periodontal ligament if the protocol is respected."

"If the working protocol for root canal irradiation is followed, no morphological changes should occur in the root walls or the surrounding tissues, as a result of the generated high temperatures."

In the work that you did, it's possible to see very high temperatures (200-400° C) inside and outside of the root. This can cause severe damage to the dentinal walls and PDL.

If this technique will be applied to small roots with a less quantity of dentine around the root canal what will happen (temperatures inside and outside of the root)?

Which temperature will reach the irrigant if it will be used? And if it will be used and it will go beyond the apex what will happen?

You should do SEM evaluation of the dentinal walls after applying this technique in order to analyze if areas of melting point will be created.

Author Response

Thank you for the evaluation and for the recommendations!

Point 1: In the work that you did, it's possible to see very high temperatures (200-400° C) inside and outside of the root. This can cause severe damage to the dentinal walls and PDL.

Response 1: This laser irradiation protocol of the root canal presented in this study is recommended by specialists as a safe and effective one. The purpose of this study was to evaluate the temperatures recorded in the dental and periodontal tissues. Although the recorded temperature values ​​are very high, we, alongside other specialists who performed similar studies (Gutknecht) consider that they cannot affect the dental and periodontal tissues because they are maintained for very short periods of time, less than 0.5 sec. In the presented study, it was observed that there were no temperature variations at the PDL or alveolar bone level and implicitly at that level there will be no thermal alteration. However, we propose in the future to carry out a SEM study that will allow the examination of the internal root wall after applying this protocol, in order to identify potential injuries.

Point 2: If this technique will be applied to small roots with a less quantity of dentine around the root canal what will happen (temperatures inside and outside of the root)?

Response 2: One weak point of our study is that the temperature analysis is done on only one tooth. It is our desire, to work on a similar study in which we can analyze the relationship between the thickness of the radicular walls and the temperature generated during the laser protocol. As previous researchers have found, the instrumentation of root canals using files with a large taper determines the reduction of the thickness of the root walls. It is expected that there is a direct correlation between the reduced thickness of the root walls and the thermal alteration of dental and periodontal tissues during laser irradiation.

Point 3: Which temperature will reach the irrigant if it will be used? And if it will be used and it will go beyond the apex what will happen?

Response 3: One future research that we want to conduct is a study to simulate the activation of the irrigation solution by irradiation with a high-power laser. This study will allow us to analyze the temperatures in the dental and periodontal tissues, but also in the inserted liquid. It is known that the action of endodontic irrigants is improved when they are heated, but it will be interesting to investigate, which is the temperature generated by laser irradiation and accepted by the human body.

Point 4: You should do SEM evaluation of the dentinal walls after applying this technique in order to analyze if areas of melting point will be created.

Response 4: A future direction of research for us is represented by an OCT and SEM study on the palatal roots of the upper molars, using the same laser irradiation protocol to observe whether thermal alterations of the internal root wall actually occur or not. In this way, the obtained results can be correlated.

Reviewer 3 Report

In the manuscript entitled “Analysis of temperatures generated during conventional laser irradiation of root canals – a finite element study” the authors aimed to determine the thermal behavior of a maxillary first molar when performing the conventional irradiation technique using a diode laser. The study is interesting and the topic well presented. Here some minor suggestions to improve the quality of the manuscript:

INTRODUCTION

_ It may be useful to mention the main mechanisms of action of EDTA and sodium hypochlorite.

_ Likewise, I suggest mentioning the effects of lasers on bacteria in the root canal.

MATERIALS AND METHODS

_ Figures 1 and 2 are grainy.

DISCUSSION

_ Describe the limitations of the study and give direction to future research.

Author Response

Thank you for the evaluation and for the recommendations!

Point 1: Introduction can be improved. It may be useful to mention the main mechanisms of action of EDTA and sodium hypochlorite. Likewise, I suggest mentioning the effects of lasers on bacteria in the root canal.

Response 1: The introduction has been improved by adding information about the way to prepare root canals, the main mechanisms of action of EDTA and sodium hypochlorite and the effects of lasers on bacteria.

Point 2: MATERIALS AND METHODS_ Figures 1 and 2 are grainy.

Response 2: Figures 1 and 2 have been improved

Point 3: DISCUSSION  - Describe the limitations of the study and give direction to future research.

Response 3: Discussions have been improved as recommended and now they also mention weak points of the study and future research directions.

Round 2

Reviewer 2 Report

In my opinion, the work can't be accepted because the irrigation technique shown can be clinically dangerous.